# Determining flow directions in river channel networks using planform morphology and topology

Jon Schwenk[1*], Anastasia Piliouras[1], Joel C. Rowland[1]

[1]Los Alamos National Laboratory, Earth and Environmental Sciences Division

*Correspondence to:* Jon Schwenk (jschwenk@lanl.gov)

**Abstract.** The abundance of global, remotely-sensed surface water observations has accelerated efforts toward characterizing and modeling how water moves across the Earth's surface through complex channel networks. In particular, deltas and braided river channel networks may contain thousands of links that route water, sediment, and nutrients across landscapes. In order to model flows through channel networks and characterize network structure, the direction of flow for each link within the network must be known. In this work, we propose a rapid, automatic, and objective method to identify flow directions for all links of a channel network using only remotely-sensed imagery and knowledge of the network's inlet and outlet locations. We designed a suite of direction-predicting algorithms (DPAs), each of which exploits a particular morphologic characteristic of the channel network to provide a prediction of a link's flow direction. DPAs were chained together to create "recipes", or algorithms that set all the flow directions of a channel network. Separate recipes were built for deltas and braided rivers and applied to seven delta and two braided river channel networks. Across all nine channel networks, the recipes' predicted flow directions agreed with expert judgement for 97% of all tested links, and most disagreements were attributed to unusual channel network topologies that can easily be accounted for by pre-seeding critical links with known flow directions. Our results highlight the (non-)universality of process-form relationships across deltas and braided rivers.

## 1. Introduction

River channel networks (CNs) sustain communities and ecosystems across the globe by delivering and distributing fluxes of water, sediment, and nutrients. Under a changing climate and widespread anthropogenic influences, modeling the transport of riverine fluxes has become vital for predicting changes in flooding hazards (Hirabayashi et al., 2013; Milly et al., 2002), habitat availability (Erős et al., 2011; Gilvear et al., 2013), contaminant transport, and water resources. CN structure affects both spatial and temporal patterns of riverine fluxes that control changes in habitat availability (Benda et al., 2004; Grant et al., 2007), flooding and floodplain nourishment (Edmonds et al., 2011), and biogeochemical cycling (Czuba et al., 2018; Hiatt et al., 2018). Flow directionality, defined as the direction of flow within each channel of a network, is critically important for vector-based models that route fluxes through CNs and are built atop a graphical representation of the CN (Czuba and Foufoula-Georgiou, 2014, 2015; Lehner and Grill, 2013). Additionally, recent research seeking to characterize deltas and braided rivers based on network structure relies on CN metrics that require knowledge of flow directions for each link (Marra et al., 2014; Tejedor et al., 2015a, 2015b, 2017).

In reality, the flow direction of river discharge may not be steady through time or may be multiple directions simultaneously. Such bi-directional flows may result from large, irreversible perturbations to the channel network (e.g. Shugar et al., 2017), fluid density differences within the channel (e.g. Garcia et al., 2006), or most commonly tidal influence (Fagherazzi et al., 2004). In these cases, local velocity measurements are usually needed to reliably

ascertain the flow direction at a given time and location. Although delta CNs often feature some tidally-influenced bidirectional channels, we focus only on the long-time, steady-state flow direction of discharge as it moves from the delta apex to its shoreline.

For watershed-scale (and larger) modeling of river tributary networks, flow directionality can often be ascertained

from knowledge of the CN structure and/or a digital elevation model (DEM) (Czuba and Foufoula-Georgiou, 2014; Dottori et al., 2016; Lehner et al., 2008). However, for dense CNs like those of a delta or braided river, a DEM may be unavailable or too coarse to characterize the flow direction of each link within the CN. Even where a DEM is available, the low slopes characterizing most deltas require high vertical precision for reliable estimates of flow directions. Additionally, both deltas and braided river CNs may be dense with short links that require high spatial

resolution elevation data to capture the elevation difference across their lengths. Even when a high-resolution DEM is available, the presence of shoals and bifurcations in multi-threaded CNs can result in flows that travel upslope, requiring sophisticated techniques to resolve flow directions (van Dijk et al., 2019; Kleinhans et al., 2017). These challenges render popular DEM-based hydrologic processing algorithms (Schwanghart and Scherler, 2014; Tarboton, 1997) and related products (Lehner et al., 2008; Yamazaki et al., 2019) ineffective. A method for estimating flow

directions of links in a CN without auxiliary data would overcome these shortcomings.

With the burgeoning availability of global remotely-sensed surface water products (Allen and Pavelsky, 2018; Pekel et al., 2016; Yamazaki et al., 2015), mapping CN morphologies has become almost trivial. However, the ease of identifying CNs is accompanied by a need for tools that can automatically abstract, model, and analyze CN imagery.

Classically, river boundaries, channel networks, and flow directions were simply resolved by hand (Bevis, 2015; Leopold and Wolman, 1957; Marra et al., 2014; Tejedor et al., 2015a), a time-consuming process subject to the operator's judgement. In this work, we present a flexible framework for automatically estimating flow directions in all links of a delta or braided river CN objectively and rapidly that requires only the CN's planform morphology and knowledge of its inlet and outlet locations. While this work focuses primarily on the techniques developed for setting

flow directions, analysis of the most effective algorithms also provides clues toward understanding how a dominant flow direction is expressed through a CN's morphologic and topologic CN characteristics.

The remainder of the paper is structured as follows: Section 2 describes the datasets used to create channel network topologies. Section 3 describes the algorithms we designed to set flow directions for all links of a CN. Section 4

assesses the accuracy of our approach, highlights where our method might fail, and discusses how particular characteristics of a river or delta's network relates to uncertainty in directionality. Improvements to reduce errors in setting link directionalities are also discussed.

## 2. Masks and Networks

We tested our method on a variety of channel networks (CNs) in order to sample a wide range of configurations and scales (Fig. 1). In particular, we selected CNs where network outlets are clustered along disparate regions of the shoreline (Niger, Yukon, Colville), where many channels flow roughly perpendicular to the apparent general flow direction (Lena, Mackenzie, Brahmaputra, Indus), where channel widths span a wide range (Kolyma, Yenisei), and where channels are heavily tidally-influenced (Niger). Only two braided river CNs were selected because braided river CNs exhibit less macro-morphologic variability than delta CNs, and the total number of braided river links we analyzed surpassed that of the deltas. The algorithms presented herein require three independent but related data: 1) binary image of the channel network, 2) vector representation (including connectivity) of the channel network, and 3) locations of inlet and outlet nodes.

The binary image of a CN, or a "mask", is simply a raster wherein "on" pixels belong to the network (Fig. 1, blue). In general, our masks include pixels identified as water or connected-to-water, unvegetated bars. Channel masks for all deltas except the Niger were created from Landsat imagery classified using eCognition software  (see Piliouras and Rowland, *in revision*). The Niger CN mask was created from the Global Surface Water monthly-integrated maps, also based on Landsat imagery (Pekel et al., 2016). Both the Brahmaputra and Indus River masks were taken from the Global River Width from Landsat mask of Earth's rivers at mean annual discharge (Allen and Pavelsky, 2018). Islands of size 20 pixels or less were removed (filled) from all channel networks. This infilling, though not strictly necessary, eliminated smaller channels that play relatively unimportant roles in the network structure. Georeferenced .tif files of the Niger, Brahmaputra, and Indus CNs are provided as Supplementary Info; other CNs are downloadable from Piliouras and Rowland, (2019).

The topology of each channel network was resolved from its mask into its constituent links and nodes (nodes shown in Fig. 1) using the Python package RivGraph (Schwenk et al., 2018). Given an input CN mask, RivGraph vectorizes the skeletonized (Zhang and Suen, 1984) mask into links and nodes and stores their connectivities. RivGraph also appends links' morphologic properties including centerline coordinates, channel width at each coordinate, average channel width, and length. RivGraph ensures that all connectivities present in the original masks are preserved in the vector representation. Finally, input and output nodes of each channel network are identified either manually or by RivGraph. Shapefiles of links and nodes and their associated properties for each CN are provided as Supplementary Info. While we used RivGraph to vectorize the network, a number of other tools are available for channel centerline extractions: RivMAP (Schwenk et al., 2017), RivWidth (Pavelsky and Smith, 2008), Rivamap (Isikdogan et al., 2017), and RivWidthCloud  (Yang et al., 2019) are among these. However, these approaches would require manual construction of the network's connectivity.

# 3. Setting Channel Flow Directions

We found no single method sufficient to accurately set all links' flow directions across the variety of tested CNs. We therefore developed a number of sub-algorithms to predict link directionality, deemed here as direction-predicting algorithms (DPAs). Each DPA falls into one of three classes: Exact, Exploitative, or Heuristic. Exact DPAs are those that enforce continuity by ensuring that flow at any point within the CN has a path to an outlet. Sources and sinks are not allowed within CNs, except for pre-identified inlets and outlets. Exploitative DPAs are those that exploit known relationships between particular morphologic or topologic features and dominant flow directions but may not hold for all links within a CN. Finally, a Heuristic DPA is one that assumes an often-intuitive rule but has no strong or formal theoretical basis. Heuristic DPAs were developed through a combination of trail-and-error and qualitative observations of many CNs. With the exception of Exact DPAs, each DPA has an associated uncertainty that is quantified uniquely depending on the particular DPA. Each uncertainty quantity may be thresholded ($\omega_{subscript}$), where *subscript* denotes the relevant uncertainty quantity. These measures of uncertainty are vital for determining flow directions of links, for example where DPAs disagree. The rationales and implementations for each DPA, along with the definitions of uncertainty, are described in Section 3.1.

By chaining together DPAs, "recipes" for setting all flow directions of a CN may be designed. Due to the qualitatively different natures of relatively confined and elongated braided river CNs compared with distributed, multi-directional delta CNs, we developed two separate recipes for fully setting CN directionality of deltas (DR) and braided rivers (BR). Similarly, delta- and braided river-specific DPAs were developed to exploit the qualitative differences between delta and braided river CNs. Section 3.3 describes how DPAs were assembled to create the BR and DR.

## 3.1. Direction-Predicting Algorithms (DPAs)

### 3.1.1 Exact DPAs (no uncertainty)

*IO*: inlets and outlets. Links attached to inlets and outlets are predicted such that flow travels away from inlet nodes and towards outlet nodes.

*PAR*: parallel links, Fig 2a. Parallel links occur when two links begin and end at the same node. To avoid creating a cycle within the graph, all parallel links must flow the same direction. As a consequence, if the direction of one of a group of parallel links is known, the others are predicted the same direction.

*CON*: continuity, Fig 2b. Enforcing continuity ensures that no sources or sinks appear within the network other than the inlet and outlet nodes. Continuity is enforced at each node by first identifying nodes where only one connected

link's direction is unknown. If the remaining group of known links are all either entering or departing the node, the unknown link is predicted to an orientation opposite the group.

*BDG*: bridge links, Fig. 2c. Bridge links are those for which all flow must travel through to reach an outlet. Removal of a bridge link from a CN breaks the connectivity of the CN, forming two disconnected CNs. Bridge links are identified in a CN graph via NetworkX's (Hagberg et al., 2008) *bridges* function and temporarily removed, creating two subnetworks. Each subnetwork is searched for the presence of inlet and outlet nodes. If either of the subnetworks has either only inlets or only outlets, the flow direction for the bridge link can be predicted as either away from the subnetwork containing the inlets or toward the subnetwork containing the outlets. In some cases, both subnetworks may contain both inlets and outlets; the bridge link direction is thus not predictable.

### 3.1.2 Exploitative DPAs

*MDC*: minimize direction change, Fig 2d. *MDC* is based on the principle that branching angles are more likely to be acute, as observed in both inland (Devauchelle et al., 2012; de Serres and DeRoy, 1990) and deltaic (Coffey and Shaw, 2017) CNs. We extend these observations by hypothesizing that the change in flow directions should be minimized at each node. Candidate links for *MDC* are identified as unknown links connected to at least one known link. At each end node of a candidate link there may be one or more links flowing into or out of the node. Each of these links, along with the candidate link, is represented by a unit vector whose direction is defined by its endpoint locations ($l_u$ for the unknown candidate link). If multiple links flow into (or out of) the node, their unit vectors are averaged to provide a single direction vector ($l_i$ and $l_o$ for into- and out of-node, respectively). The goal is to determine which of $l_i$ or $l_o$ is most parallel to $l_u$; thus angles are computed between $l_i$ , $l_o$ and and $l_{u0}$, $l_{u1}$, where $l_{u0}$ represents the original position of the unknown link, and $l_{u1}$ represents its 180° rotation about the node. The minimum of all angles is computed via Eq. 1:

$$\alpha_{min} = \min(\alpha_{u_o,l_i}, \alpha_{u_o,l_o}, \alpha_{u_i,l_i}, \alpha_{u_i,l_o}) , \tag{1}$$

where the subscripts denote the vectors defining the angle. If $\alpha_{min} = \alpha_{u_o,l_o}$ or $\alpha_{u_i,l_i}$, the unknown link is set to flow out of the node, else into it. Where possible, this procedure is repeated for both end nodes of $l_u$, and $\alpha_{min}$ becomes the minimum of both nodes' minima. The magnitude of $\alpha_{min}$ provides a measure of certainty of the prediction; $\alpha_{min}$ closest to 0 represent links whose flow directions are more aligned with at least one of the known connected links. A threshold ($\omega_{ang}$) may thus be set on $\alpha_{min}$ to specify the maximum level of direction change allowed before setting the unknown link's direction.

*SDEM*: synthetic DEM (deltas only), Fig. 2e. As discussed in Section 1, DEMs may provide valuable information toward discerning flow directions, but are often too coarse for use with low-sloped delta CNs. *SDEM* invokes our conceptualization of long-time, steady-state flow that moves from the apex of a delta to its outlets to construct a

synthetic DEM. This procedure creates inlet and outlet DEMs separately, designed such that elevations are higher near inlets and lower near outlets. The final synthetic DEM is simply the sum of the inlet and outlet DEMS.

For the outlet DEM, an image of the same size and resolution of the input mask is created and filled with ones. To estimate the delta's shoreline, the convex hull of the outlet nodes is computed, and the edge of the convex hull connecting the two most distant outlet nodes is removed to provide an ordered set of input nodes. Line segments between each input node are linearly interpolated at 0.1 pixel intervals, and this interpolated shoreline is "burned" into the image of ones by lowering their elevations to zero. A distance transform (Jones et al., 2001) of the image returns an image where each pixel's value represents its distance to the nearest shoreline. This image ($I_{DEM,o}$) is normalized on the interval [0, 1] according to

$$I_{DEM,o} = \frac{I_{DEM,o} - \min(I_{DEM,o})}{\max(I_{DEM,o}) - \min(I_{DEM,o})}. \tag{2}$$

The inlet DEM ($I_{DEM,i}$) is constructed similarly, but with some exceptions. Only inlet nodes whose associated channel widths are at least 75% of the widest inlet channel are considered. Before normalization (Eq. 2), $I_{DEM,i}$ is inverted via

$$I_{DEM,i} = \max(I_{DEM,i}) - I_{DEM,i} \tag{3}$$

to ensure that elevations near the inlets are raised rather than lowered. The final synthetic DEM is simply the sum of $I_{DEM,o}$ and $I_{DEM,i}$. The synthetic DEM for the Mackenzie Delta is shown in Fig. 2e; only one of the inlet nodes contributed to its $I_{DEM,i}$.

The slope of each link may be computed by drawing elevation values from $I_{DEM}$, and a prediction of a link's flow direction can be made. Channels often flow perpendicular to the general flow direction dictated by $I_{DEM}$, so predictions made by **SDEM** may be poor. However, the magnitudes of a link's slope and its length serve as measures of certainty; links may be thresholded by length ($\omega_{len}$), slope ($\omega_{slope}$), or both to ensure that **SDEM** only sets the longest, steepest links.

**MC**: main channels, Fig 2f. Typically, but with exceptions, the main channels (i.e. those that transport the largest discharge) of a braided river or delta CN are the widest of the CN. This concept originates in well-studied, quasi-universal hydraulic geometry relationships of the form $W=aQ^b$ for width ($W$), discharge ($Q$), and fitted parameters ($a,b$) (Leopold and Maddock, 1953; Parker et al., 2007). We impose two additional constraints to this relationship to define main channels: they must begin at inlets and end at outlets, and they tend to follow the most direct path. Under these conditions, each outlet has a corresponding "shortest and widest" path from each inlet. This path is found by creating a weighted graph of the CN, where weights are defined according to

$$wt_i = l_i \cdot \left( \max(w) - w_i \right) \qquad\qquad\qquad (4)$$

for the $i^{th}$ link with length $l_i$ and width $w_i$. This weighting scheme results in larger weights for longer and narrower channels. The shortest paths of the weighted graph are computed from each inlet to each outlet using Djiktstra's Method implemented in NetworkX. The direction of each link along each path may then be predicted according to the ordered list of nodes returned. A CN may contain multiple main channels; if a link's direction has already been predicted by a main channel, it is not re-predicted by other main channels that share it. Therefore, in rare cases where

two main channels might predict opposite flow directions for a link, the link is predicted by only the flow direction of the first. No uncertainty measurements are made for **MC**.

**VD**: valleyline distance (braided rivers only), Fig 2g. Rivers typically flow through corridors of some sort, referred to here as valleys. Valleys feature the lowest elevations in a landscape and contain river floodplains. Multi-scale

analyses of river valleys indicate that significant information about the local (link) scale is shared with the valley scale (Gutierrez and Abad, 2014; Vermeulen et al., 2016). **VD** attempts to impart flow direction information from the river valley-scale to the link-scale.

A river corridor centerline is created by filling the holes in the CN mask, skeletonizing it, and smoothing. A mesh is

generated over the CN by drawing perpendicular line segments along the centerline. This mesh-generation technique was introduced by (Schwenk et al., 2017) and adapted to a Python implementation here. Knowledge of the inlet and outlet nodes' locations allows an ordering of the polygons and perpendiculars comprising the mesh.

A prediction for each link is made by finding the two perpendiculars that encompass the link's endpoints (dotted

white lines, Fig. 2g). The link's upstream node is predicted as the one closer to the upstream perpendicular. Similarly to deltas, channels of a braided river may flow approximately perpendicularly to the centerline, resulting in an uncertain prediction. To account for the certainty of **VD**, the number of perpendiculars required to encompass a link ($N_{perps}$) is also computed. Links passing through more perpendiculars carry a greater prediction certainty, and a threshold ($\omega_{perps}$) may be applied to predict only the most certain links.

**VA**: centerline angle (braided rivers only), Fig 2g. The logic behind **VA** exactly follows that of **VD**, but instead of considering downvalley distance, we consider the local angle of the valley centerline. Flow direction can be predicted by comparing a link's angles with the nearby centerline angle. The endpoints of the link are mapped to the nearest perpendicular, and the centerline angle between these two perpendiculars ($a_{cl}$) is computed (Fig. 2g). The angles of

the link computed from the vector defined by its endpoints ($a_0$) and its 180-degree rotated version ($a_1$) are also computed. The link's direction is predicted as the orientation whose angle is closest to $a_{cl}$. The difference between $a_{cl}$ and the closer of $a_0$ and $a_1$ provides a measure of certainty of **VA**, with smaller differences corresponding to higher

certainties. This difference may be thresholded ($\omega_{cla}$) to specify the level of parallelism between the link and the valley centerline required to make a prediction.

### 3.1.3 Heuristic DPAs

**SP**: shortest path. **SP** stemmed from the observation that in most cases where flow direction is unknown, the true flow path corresponded to the shortest distance to its outlet. The **SP** implementation is identical to **MC** except the links are unweighted. In cases where the shortest path between inlets and outlets results in opposite predictions of flow direction for a link, the mode is selected as the prediction. **SP** may fail when macro-morphology of the CN, e.g. the change in Brahmaputra's valley direction from west to south (Fig. 1), imposes a low-frequency direction change.

**PMC**: parallel to main channel, Fig 2h. Similarly to how **VA** and **VD** transfer information from the valleyline to predict individual links, the links of main channels contain information of local flow directions that may be exploited to predict nearby, approximately-parallel links whose flow directions are unknown. For each link that is not part of a main channel, the nearest (Euclidean distance) main channel node is found. Each of the endpoint nodes of the unknown link is mapped to their nearest main channel nodes (for example, $d_1$ and $d_2$ in Fig. 2h). If the endpoint nodes map to the same main channel node, no prediction can be made for the link. In all other cases, a prediction can be made that aligns the flow direction of the unknown link with the main channel nodes to which its end nodes were mapped. The strength of this prediction ($\omega_{nodes}$) is captured by the difference of mapped-to node positions along the main channel. In Fig. 2h, for example, this number is one ($\omega_{nodes} = MC_2 - MC_1$).

**MMA**: multiple methods agree. If DPAs disagree about the flow direction of a link, **MMA** simply chooses the most common prediction. A minimum number of agreeing DPAs may be enforced ($\omega_{agree}$) to ensure greater certainty of predictions made by **MMA**.

## 3.2. Recipes for Deltas and Braided Rivers

DPAs provide a number of tools for predicting flow directions, and they may be assembled into "recipes" designed to set flow directions for all links in a CN. Morphologic variability across our study deltas and braided rivers prevented the design of a "one size fits all" recipe, so we designed both a delta recipe (DR) and a braided river recipe (BR). The arrangement of and thresholds applied to the DPAs used to construct each recipe are detailed in Fig. 3; here, the guiding design principles are discussed.

DPAs provide predictions of some link directions, and each prediction has an associated uncertainty. Only **IO**, **CON**, **PAR**, and **BDG** are fully deterministic (i.e. unreliant on thresholding), while all other DPAs provide predictions based on some degree of thresholding. Because some DPAs are only effective when some links' directions are already known (i.e. **MDC** and **MAA**), a recipe must be designed that sets links iteratively, rather than all-at-once.

Setting links iteratively is disadvantageous because an improperly-set link's direction may "infect" nearby links (i.e. cause them to be improperly set), and the infection may spread throughout the network. However, an iterative approach also allows links to be set from most-certain to least, minimizing the likelihood of an infection.

Most DPAs provide a metric of uncertainty in addition to their prediction. By applying thresholds ($\omega$) to these metrics, directions may be set for only a DPA's most certain links rather than applying the DPA to all links. For example, for **SDEM**, longer and steeper links are more certain, so the first call to **SDEM** in the DR is only applied to links that are in the upper-25[th] and upper-50[th] percentiles for length and slope, respectively (i.e. $\omega_{len}$=25% and $\omega_{slope}$=50%). Thresholds for DPAs also include $\omega_{ang}$ (**MDC**), $\omega_{nodes}$ (**PMC**), $\omega_{agree}$ (**MAA**), $\omega_{n\_perps}$ (**VD**), and $\omega_{cl\_ang}$ (**VA**). For the angle-based thresholds, smaller values correspond to higher certainty, and conversely for the non-angle-based thresholds. The meaning of these thresholds was described in detail in Section 3.1.

The most certain non-Exact DPAs are those containing information of the general flow direction--**SDEM** for deltas and **VD** and **VA** for braided rivers. These are applied second, following the Exact DPAs. Continuity (**CON**) is not explicitly shown in the recipes (Fig. 3), but whenever a link's direction is set, all its connected links are attempted to be set by **CON**. Each time **MDC** is applied, the threshold $\omega_{ang}$ is applied in equally-spaced intervals of 10 to ensure most certain links are set first. For example, $\omega_{ang}$=1.0 would apply **MDC** with $\omega_{ang}$= (0.1, 0.2, … 1.0). It is possible that the BR fails to set all links' directionality; however, we found through visual inspection that flow directions of these unset links were ambiguous, and their flow directions are thus set randomly. Similar links exist in delta CNs, but **SDEM** is used to set their directions in the DR. Attempts to fix internal sources/sinks and cycles are made at the ends of both the DR and BR.

## 3.3. Cycles and Continuity

After all link directions of a CN have been set, the resulting graph may contain interior sources or sinks and/or cycles. A cycle is a set of directed links and nodes for which a node is reachable from itself. While it is possible that a real CN may truly contain a cycle, our conceptualization of a CN as delivering all fluxes from its apex(s) to its outlet(s) precludes their existence in our graphs. Thus cycles identified in a CN indicate a set of links for which at least one link flows in a direction opposite of what is desired; in other words, cycles identify links that should be corrected. Cycles are identified via the NetworkX method *simple_cycles()*. Sources and sinks are identified by ensuring that for all interior nodes (i.e. not inlets or outlets), at least one link departs the node and one link enters it.

If an interior source or sink is present in a CN, a "fix" is attempted. Its goals are to flip the directionality of a single link so that a) the source/sink is no longer present, b) the flipped link does not create another source or sink, and c) the flipped link does not create a cycle. To fix the source/sink, each link connected to the source/sink node is flipped and continuity is re-evaluated. If the link violates continuity post-flip, it is discarded from consideration. For each of the flipped links that did not violate continuity, if flipping its direction creates a cycle, it is also discarded. If more

than one links meet these criterion, the shortest link is selected as the one to be fixed (flipped), as DPAs are generally more certain about longer links.

Cycles may be more complicated to fix automatically because there is no upper bound on the number of links they may contain. In practice, cycles typically contained fewer than ~10 links, so an automated cycle fix was implemented. This procedure simply unsets all the directions of links in a cycle, with the exceptions of directions that were set via *IO*, *MC*, *BDG*, *SDEM*, *VD*, or *VA*. The unset links are then reset according only to *MDC*, beginning with the most-certain angles (lowest $\omega_{ang}$) and longest links. After all links have been reset, a check ensures the cycle

has been resolved. If the cycle persists, the same procedure is repeated except the directions of the cycle links plus all links connected to the cycle are initially un-set. If the cycle still remains unfixed, links are returned to their original directions and the cycle is noted for manual inspection.

## 3.4. Validating Flow Directions

In the absence of data for all links of all CNs that would allow a deterministic evaluation of each link's flow

direction, we created a validation database of link directions set according to the judgements of a delta and a braided river expert. For each CN, at least 10% of the total number of links were randomly selected, and their directions were determined manually by the experts using only the same information available to the recipes, i.e. the channel network mask and its graph. Each of the selected link ids were stored along with the experts' best judgement of the corresponding upstream node id. We note that the recipes were developed prior to the development of this validation

database. Each disagreement between the expert and the recipe-predicted link direction was investigated, and we also counted the number of expert-errors either due to mistaken data entry or obviously incorrect judgement; expert errors were less than 4% across all individual CNs with an average of 1.7% for all sampled links (Table 1).

# 4. Results and Discussion

## 4.1. Overall Accuracy of the Recipes

Overall, we found 97.0% and 98.2% agreement between expert judgement and links set according to the DR and BR, respectively. Henceforth, we consider expert judgement to be "the truth" and refer to disagreements as errors, although the expert judgements were also subject to mistakes (Section 3.4, Table 1). No errors were found within four of the seven delta CNs, with the Niger CN having the highest error rate (9.5%) followed by the Mackenzie (5.3%) and Lena (3.4%) CNs. The BR performed similarly for both braided river CNs, with errors of 2.3% and 2.2%

for the Brahmaputra and Indus CNs, respectively. No CNs contained internal sinks or sources, but 4/9 CNs did contain cycles. Of these, only a single cycle was not automatically resolvable for the Lena and Brahmaputra CNs.

## 4.2. Erroneous links

Each of the 42 identified links that were erroneously set by our recipes was inspected to identify where and how DPAs are likely to fail. Due to the iterative nature of the recipes, erroneous links set early in a recipe are more likely to infect neighboring links, and we found that erroneous links were rarely isolated but occurred in clusters. Because of this, evaluating the accuracy of a particular DPA requires deeper investigation than simply counting the number of erroneous links set by that particular DPA. For example, if *MC* erroneously sets a link, *MDC* may use the local flow direction of the mis-set link to erroneously set further links.

The following subsections describe sources of errors, including the most common error (Section 4.3.1) and morphologic properties of the Lena (Section 4.3.2) and Niger (Section 4.3.3) CNs that were problematic for our recipes. These subsections explain 29/42 of the identified erroneous links. Of the 13 remaining errors, *PMC* was responsible for two, *MAA* was responsible for one, and *MDC* was responsible for 10. We note that not all erroneous links were identified in the CNs as we only tested >=10% of the total links in each CN. However, because errors tended to occur in clusters and links were randomly sampled for testing it is likely that we captured all the major sources of errors.

### 4.2.1. Ambiguous links

Sixteen of the 42 link direction errors were attributable to ambiguous links for which morphology alone cannot provide certainty of flow direction. Generally, ambiguous links flow perpendicularly to the local (or overall) flow direction (Figs. 5a-d). Flow directions through ambiguous links can reasonably be argued to go both directions, and in many cases bi-directional flow may be reality (e.g. Fig. 5a). In these cases, *MDC* cannot be applied with certainty due to the high junction angles; nor can *SDEM* be applied with certainty because ambiguous links are typically short and not parallel to the main flow direction. *MAA*, which employs shortest-path methods, sets many of these links, but we found shortest path to be unreliable for CNs with large-scale morphologic variability, e.g. the ~90° bend in the Brahmaputra CN. In the case of Fig. 5c, neither *VA* nor *VD* could set the erroneous links because of their perpendicular orientation with respect to the centerline. Figure 5d shows an unusual ambiguous link created by the formation of an oxbow lake; the expert judgement was based on the flow direction before the oxbow lake was cut off from the main channel, but the modern topology suggests flow could travel in the opposite direction. We were unable to design DPAs that set ambiguous links with certainty; however, ambiguous links were the last ones (i.e. least certain) to be set by our recipes, which limited the influence that their potentially-erroneous flow directions propagated to other links in the network. Although not strictly true, ambiguous links typically play unimportant roles in overall CN routing.

### 4.2.2. The Niger CN

At 9.5%, the Niger CN contained the highest fraction of erroneous links (Table 1). However, we found that all four erroneous links shared the same source of error. The Niger delta features a number of tidal channels that are typically

wider at their outlets and eventually fade away toward the delta's apex. Some of these tidal channels are connected to the CN, while others terminate on the delta plain without a surface connection (Fig. 5e). The erroneous links of the Niger were feeder links from the main CN to a tidal channel that, while connected, likely receives very little flow from main subnetwork. In other words, fluxes to the outlet of this tidal channel should originate at the tidal channel

inlets, but these inlets were not considered to be inlets of the CN. Their absence resulted necessarily in a main channel from the CN inlet to the tidal channel outlet, which in turn forced flows right-ward towards the tidal channel and resulted in erroneous links. We verified that placing a single inlet at the source of the tidal channel resolved these erroneous links, resulting in a 0% error for the tested links of the Niger CN.

### 4.2.3. The Lena CN

The Lena CN had a total of 15 identified erroneous links and an unresolved cycle. Nine of these links and the cycle are attributed to the Lena CN's unusual structure. The Lena CN features two clusters of outlets; a long, continuous shoreline on its upper-right side contains the majority of the outlets, but a separate subnetwork delivers fluxes to the left side (directions with respect to orientation in Figs. 5f-h). Fluxes entering its inlet node are either immediately routed to the left subnetwork or flow upwards to a pseudo-apex (Fig. 5f). While the majority of flow through the

pseudo-apex heads toward the right shoreline, some is routed through smaller channels to the left shoreline. Recall that *MC* finds the shortest, widest path from inlets to outlets as a main channel. The main channel from the inlet to the outlet denoted in Fig. 5f is incorrect, as flow to that outlet node should travel through the pseudo-apex. However, because of the narrowness of the channels connecting the pseudo-apex to the outlet, the shortest, widest path bypasses the pseudo-apex and flow to the outlet approaches from the wrong side. *MC* thus erroneously sets a number

of links, including one particularly critical link (Fig. 5g). This link is critical because it bridges two subnetworks; incorrectly setting its flow direction prevents flow from the upper subnetwork from reaching the leftmost outlets. As *MC* is applied early in the DR, its incorrect direction more readily infects nearby links as evidenced by the numerous erroneous links surrounding it. Its incorrect direction also created the unresolvable cycle in the Lena; this cycle was not present when we pre-set the critical link to the correct flow direction and then applied the DR.

Four of the Lena CN's 15 identified erroneous links were attributable to the difficulty of creating a representative synthetic DEM. Elevations in the synthetic DEM are proportional to their distance from the outlet nodes and inversely proportional to the distance from the inlet nodes; this scheme created a ridge in the synthetic DEM (Fig. 5h) that divided the inlet subnetwork from the rest of the delta, effectively forcing the links of the inlet subnetwork to

flow uphill and resulting in the four erroneous links. Because elevations near the inlet are raised, the slopes of the inlet subnetwork links were relatively smaller, allowing the DR to pass over them (i.e. not set their directions) in the early stages of the recipe. Other DPAs were thus employed to set the vast majority of the inlet subnetwork links, preventing the ridge from adversely affecting their directions. Interestingly, the end of the ridge coincides with the location of the pseudo-apex due to the radial layout of the Lena's outlets.

## 4.3. Effectiveness of DPAs

Channel networks can exhibit a wide range of morphologic variability, but also contain consistent features that may be exploited by various DPAs. In order to understand which features are more universally consistent, we measured the effectiveness of a DPA by the fraction of a CN's links it sets (Fig. 6). Morphologically, delta and braided river CNs have three key differences exploitable to predict flow directions through their links. First, deltas typically have more outlets than braided rivers. *IO* reflects this difference; 5% of delta CN links were set by *IO* compared with <1% for braided rivers (Fig. 6). *SDEM* also takes advantage of the additional outlets of delta CNs to construct the synthetic DEM, accounting for setting 10% of delta CN links' directions. *MC* best exploits the delta CNs' relative numerous outlets as it finds a "main channel" from each inlet to each outlet (25% set, Fig. 6b). However, we found *MC* unsuccessful for braided river CNs because link widths were too similar to confidently define a main channel, highlighting the second morphologic difference between delta and braided river CNs: the tendency for delta main channels to be wider relative to the full width distribution, and thus more certainly identified. The average coefficient of variation of link widths reflects this; 1.05 for delta CNs compared with 0.83 for braided river CNs. Finally, the third exploitable morphologic difference between braided rivers and deltas, evident from Fig. 1, is the relatively elongated and confined domain occupied by braided rivers compared with the radial shape of most deltas. The confinement of braided rivers to a relatively narrow band permits a meaningful centerline to be resolved, which we exploited through *VD*, *VA*, and *VAD*. These three DPAs accounted for setting 54% of link directions in the braided river CNs (Fig. 6b).

Across all CNs, more than half of all links were set by *MDC* (33%) and *CON* (27%) (Fig. 6b). Thus 60% of all links were set with only local flow direction information, highlighting the importance of the accuracy of other DPAs. The basis for developing *MDC* lies in theoretical and empirical observations that indicate that channel bifurcation angles tend to deviate an average of ~36° with respect to the upstream channel direction for both river (Devauchelle et al., 2012) and delta (Coffey and Shaw, 2017) CNs. However, the distribution of bifurcation angles may be quite broad and outliers are not uncommon. *MDC* was thus applied iteratively and thresholded to be applied only to the links nearest the mean of the bifurcation angle distribution. Despite this iterative approach, *MDC* was responsible for the greatest proportion of erroneous links, highlighting the spread around the cited ~36° average. Nevertheless, the similarity of effectiveness of *MDC* across both delta and braided river CNs (Fig. 6b) suggests similar local processes at work to form and maintain channel bifurcations in both deltas and braided rivers. Although deltas and braided river CNs are shaped by shared fundamental processes of fluvial erosion and sediment transport, deltas are subject to additional processes including tides, waves, coastal currents, sea level rise, and subsidence that result in more topologic and morphologic complexity. This complexity was reflected in the construction of our recipes, which may be considered as a minimum number of rules required to accurately predict all flow directions in a CN. The DR required 15 DPA applications compared with the BR's seven.

DPAs showed fairly consistent effectiveness across the delta and braided river CNs (Fig. 6a). As expected, *MC* and *IO* were more effective for delta CNs with many outlets and fewer links (Colville, Kolyma, and Yukon). *CON* was relatively more effective and *MDC* less effective for the Niger CN than the other deltaic CNs, reflecting its smaller proportion of laterally-flowing (relative to general flow direction) links. The braided river CNs, on the other hand, feature numerous laterally-flowing links and the higher effectiveness of *CON* and *MDC* relative to the delta CNs reflects this morphologic difference in CNs. Interestingly, almost no bridge links were present in the braided river CNs, while 3% of delta CN links were set by *BDG*..

## 4.4. Improvements and Speed

With an overall accuracy of 97.7%, our recipes can provide a suitable starting point for resolving flow directions in all links of delta and braided river CNs. However, some applications (e.g. flux routing) may require complete accuracy. In these cases, perhaps the simplest and most effective method to improve accuracy is to pre-seed the CN with known flow directions. This may be done prior to the initial application of a recipe or in an iterative fashion by identifying critical, erroneous links after the recipe's application. For example, when the correct flow direction was assigned to the critical link of the Lena (Fig. 5g) before applying the recipe, 20 erroneous links and the unresolvable cycle did not occur. Fully specifying all inlet and outlet nodes is also important to improve accuracy, as evidenced by the elimination of all erroneous links from the Niger by adding a single inlet. The flexibility of the recipes allows for easy implementation of other DPAs that can be designed to exploit other morphologic CN properties and improve overall recipe performance.

*MDC* was responsible for the greatest proportion of erroneous links. Our implementation considered only the link endpoint locations to compute flow direction vectors, but in the cases of longer links, this approach may fail. An alternative and potential improvement might consider only the local link directions (i.e. the pixels of the link closest to the node), although we found this approach challenging to implement. Often the near-node segments of links do not represent a link's actual direction, especially for narrower links connected to wider ones.

In our Python implementation, runtimes for the unparallelized recipes were on the order of one second on a typical desktop machine. However, depending on the size and resolution of the underlying mask, the image processing techniques to create the synthetic DEM for delta CNs and the centerline mesh for braided river CNs can require tens of seconds. These processes must only be run once, though, allowing for rapid development and testing of other DPAs and recipes.

## 5. Conclusions

This work presents a framework for building algorithmic recipes to automatically and objectively set the steady-state flow directions in all links of a channel network (CN) graph using only a binary mask of the channel network.

Twelve direction predicting algorithms (DPAs) were presented that exploit morphologic and topologic features of a CN to predict the direction of flow within links. By chaining DPAs together, we created recipes for delta CNs and braided river CNs that set all flow directions within the CN.

Knowing only the channel network mask and the locations of inlets and outlets, our recipes for setting link directions agreed with expert opinion for 97% (delta CNs) and 98% (braided river CNs) of links analyzed. Analysis of the links that disagreed showed that special attention must be taken to design recipes for CNs with unusual morphologic features. We also found that CNs may contain critical links that, if set incorrectly, may result in many other mis-set links and cycles in the CN. However, pre-seeding the CN with the correct directions of critical links effectively "cures" such infections.

Even across the wide range of delta morphologies we examined, only a handful of DPAs were required to set the vast majority of links of the CNs. Locally minimizing the change of flow direction between links and enforcing continuity were sufficient to set 60% of links' flow directions in both delta and braided river CNs. Most of the remaining 40% were set by incorporating information from the macroscale CN by identifying main channels, constructing a synthetic DEM (deltas), or leveraging an along-centerline mesh (braided rivers). The effectiveness of *MDC* for both deltas and braided rivers points toward the dominant expression of process-form relationships in fluvial systems under a range of environmental conditions. This expression was present but more obscure in delta CNs that are affected by tidal processes.

Although we analyzed large CNs whose masks originated from 30-m resolution Landsat imagery, our recipes are generally applicable to CNs of any scale. The accuracy of our delta recipe across a broad range of delta morphologies suggests a robustness to delta CN forms and suggests that our recipes are applicable to experimental and modeled CNs as well. However, globally, CNs exhibit a wider range of morphologies and topologies than we captured in our test set. If our recipes perform poorly on other CNs, their flexibility and adaptability allow for modification to rearrange the order of DPA application, change the DPA thresholds, or incorporate new DPAs. Relative to deltas, braided rivers exhibit less macro-morphologic variability, so we expect the braided river recipe to be more generally applicable. Our framework is also applicable to other networks and network-based models where directionality is crucial to understand transport, such as in the vascular systems of plants and animals, transportation systems, and utility grids, although application-specific DPAs may need to be developed for these systems.

## Code and data availability

The algorithms and recipes detailed here are being implemented into RivGraph (Schwenk et al., 2018), a Python package for analyzing morphologies and topologies of channel networks. An unofficial release of this code can be found at https://github.com/jonschwenk/RivGraph. Georeferenced binary channel masks, distance transforms, link

directions geotiffs, and shapefiles of each channel network's directed links and nodes are provided as Supplementary Data.

## Author contribution

JS designed and implemented the method presented herein. AP was the delta expert who created the validation dataset. JS did the same for the braided rivers. JS, AP, and JR each contributed to preparing the manuscript.

## Competing interests

The authors declare that they have no conflict of interest.

## Acknowledgements

Research presented in this article was supported by the Laboratory Directed Research and Development program of Los Alamos National Laboratory under project number 20170668PRD1. This research was also funded as part of the HiLAT project through the Department of Energy, Office of Science, Biological and Environmental Research Program's Regional and Global Model Analysis program. Additionally, this work was partly supported by the DOE Office of Science, BER under the Subsurface and Biogeochemical Research Program Early Career Award to JC Rowland.

# Figures

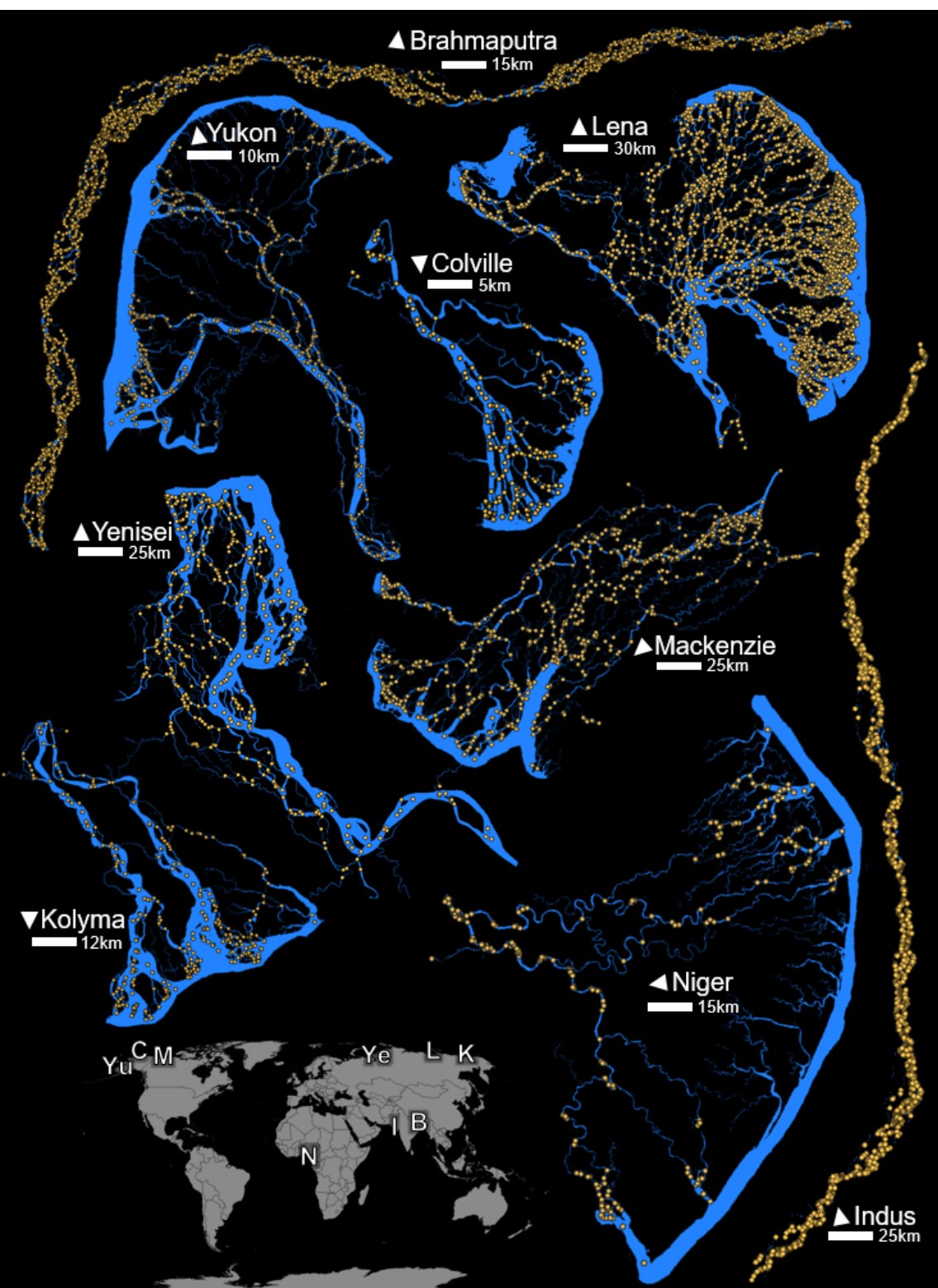

**Figure 1. Study channel networks. Channel masks are blue and nodes of the extracted networks are orange. Delta channel masks also include a portion of the ocean along the delta front. For clarity, links are not displayed but may be assumed between each pair of adjacent nodes. Locations of the (Yu)kon, (C)olville, (M)ackenzie, (Ye)nisei, (L)ena, (K)olyma, and (N)iger Deltas and the (B)rahmaputra and (I)ndus Rivers are shown on the map of continents. Arrows for each point north.**

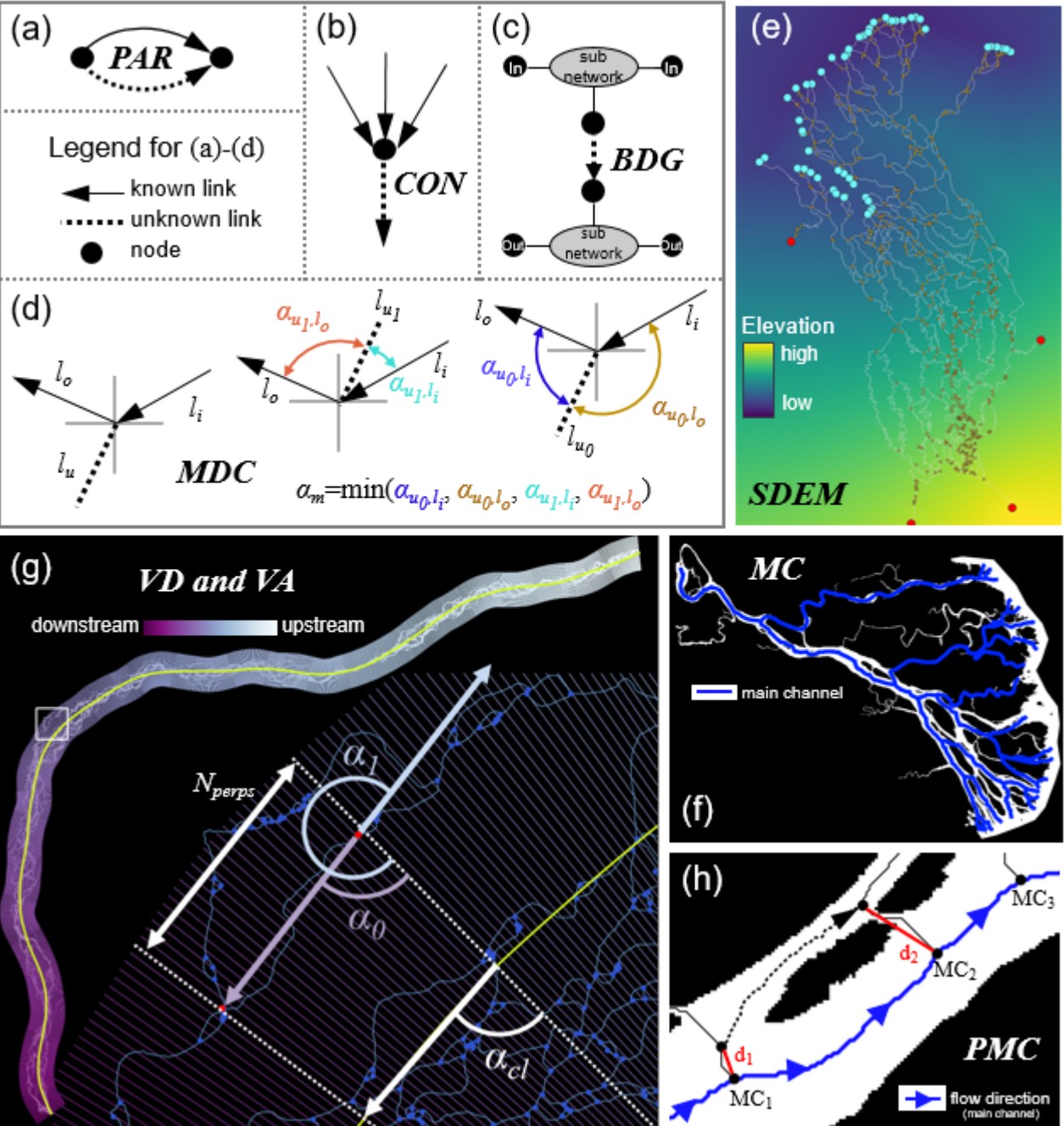

**Figure 2. Diagrams showing the direction-predicting algorithms (DPAs). Symbology is further explained in Section 3.1. (a) Predicting an unknown parallel link. (b) An example of applying *CON* to determine the unknown link. (c) Predicting a bridge link with *BDG*. (d) Using the minimum direction change *MDC* for predicting the unknown link. (*e*) A synthetic DEM (*SDEM*) for the Mackenzie Delta is shown with blue outlets and red inlets. (f) A centerline mesh is shown for the Brahamputra River with a yellow centerline to demonstrate *VD and VA*. The box denotes the bounds of the zoom-view. The unknown link's endpoints are marked by red points. Dashed lines follow the mesh perpendiculars. (g) Main channels found by *MC* for the Colville Delta are denoted by blue lines. (h) An example of main channel parallels (*PMC*). Distances $d_1$ and $d_2$ are defined in Section 3.1; $MC_n$ refers to the $n^{th}$ main channel node, ordered from upstream to downstream. The dashed link's flow direction is unknown.**

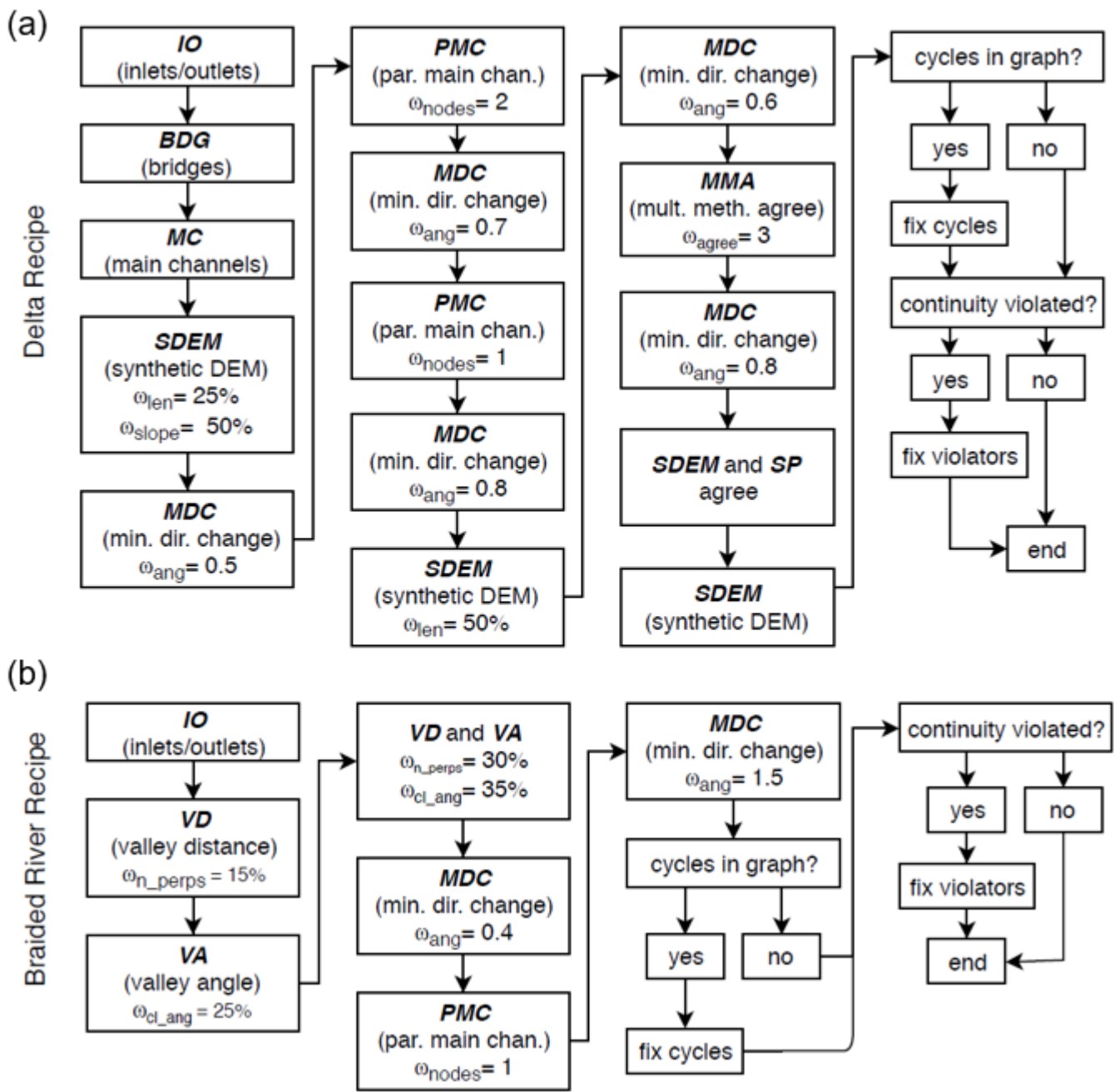

Figure 3. Recipes for setting link flow directions by chaining together DPAs. (a) Delta recipe. (b) Braided river recipe. Continuity (*CON*) is not explicitly represented in the diagrams, but is applied locally after any link's direction is set. Thresholds (ω) are implemented to ensure that only the most certain links are set by each DPA and are defined in Section 3.1. Each threshold has a different meaning that corresponds to the particular DPA.

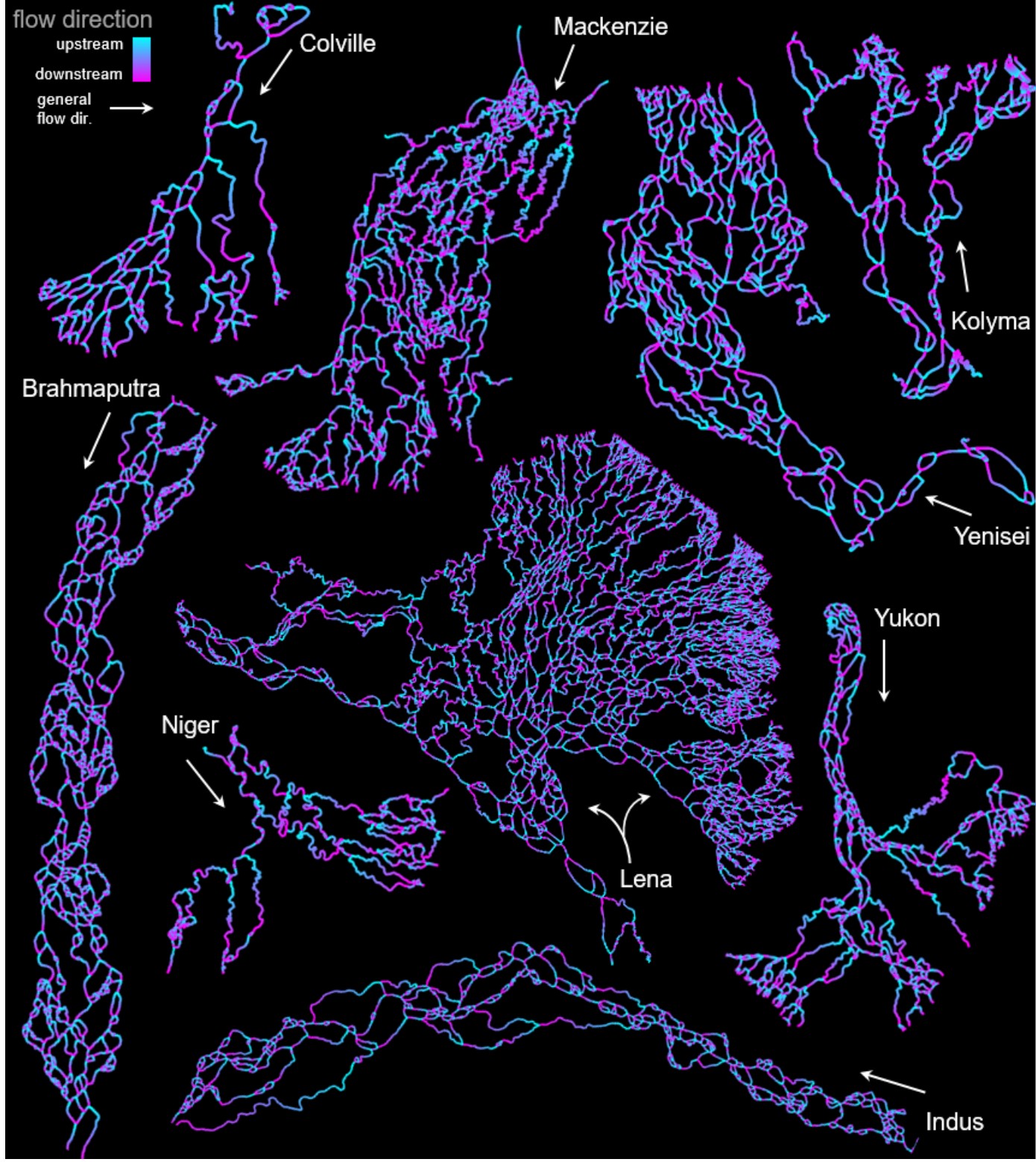

Figure 4. Flow directions for each channel network. The Brahmaputra and Indus Rivers are cropped for improved visibility. White arrows denote the general flow direction for each CN.

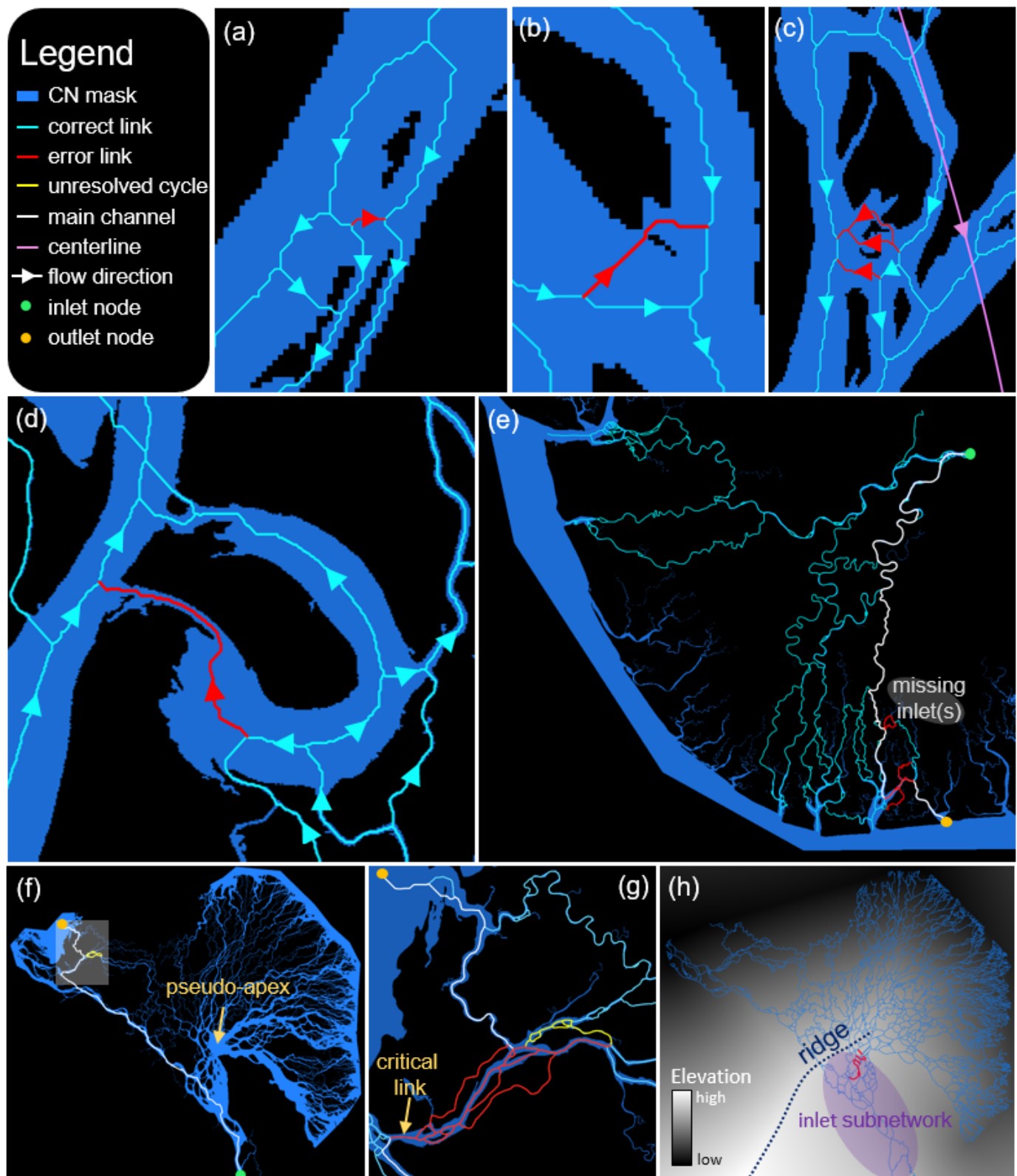

**Figure 5. Errors of the recipes. (a-d) Ambiguous links that were erroneously set for the Mackenzie (a and d), Indus (b) and Brahmaputra (c) CNs. (e) The Niger CN features tidal channels whose inlets were not considered. A main channel is shown to an outlet node that should be fed by the missing inlets. (f) A problematic main channel (white) is shown for the Lena CN. A zoom view of the shaded area is shown in (g). (h) Synthetic DEM for the Lena CN with a ridge of the synthetic DEM marked.**

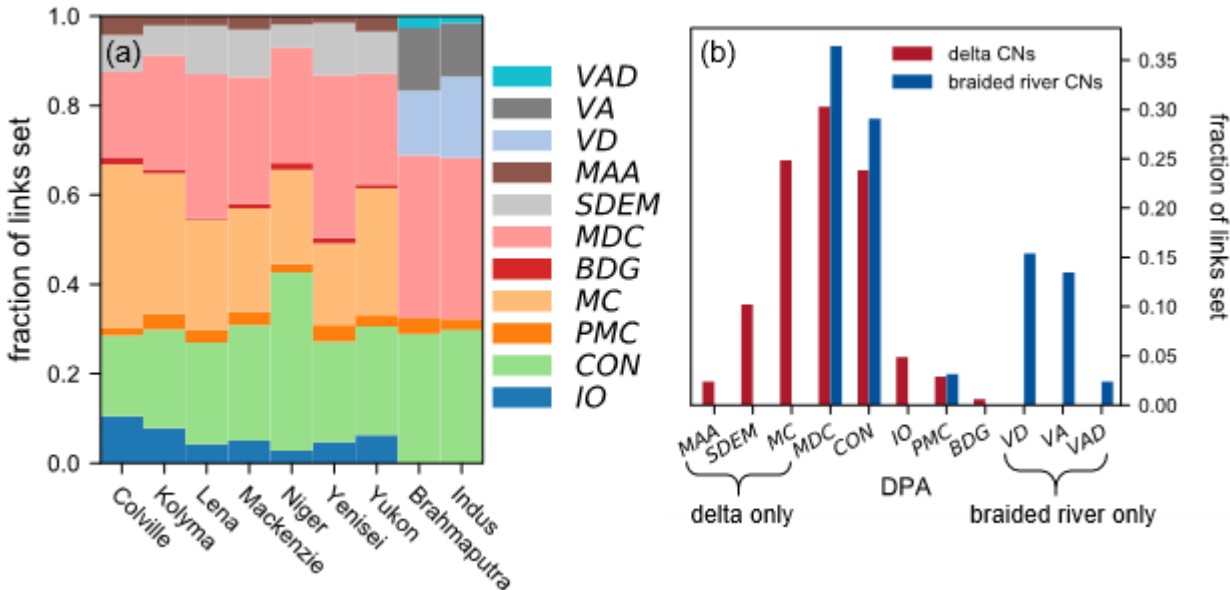

**Figure 6. DPA effectiveness. (a) Fraction of links set by each DPA for each study CN. DPAs are defined as VAD-valley angle and distance, VA-valley angle, VD-valley distance, MAA-multiple DPAs agree, SDEM-synthetic DEM, MDC-minimize direction change, BDG-bridge links, MC-main channels, PMC-parallels to main channels, CON-continuity, and IO-inlet/outlet links. (b) Fraction of each DPA for all CNs; red (and blue) bars sum to one.**

## Tables

| CN | links | cycles | cycles fixed | links compared, (%) | disagree, (%) | expert errors, (%) |
|---|---|---|---|---|---|---|
| Colville | 256 | 0 | 0 | 30 (11.7) | 0 (0.0) | 0 |
| Kolyma | 421 | 0 | 0 | 49 (11.6) | 0 (0.0) | 0 |
| Lena | 4592 | 4 | 3 | 467 (10.2) | 15 (3.2) | 2 (0.4) |
| Mackenzie | 1158 | 1 | 1 | 119 (10.3) | 6 (5.3) | 2 (1.7) |
| Niger | 365 | 0 | 0 | 42 (11.5) | 4 (9.5) | 1 (2.4) |
| Yenisei | 685 | 0 | 0 | 69 (10.1) | 0 (0.0) | 0 |
| Yukon | 750 | 1 | 1 | 80 (16.6) | 0 (0.0) | 2 (2.5) |
| Brahmaputra | 6446 | 5 | 4 | 667 (10.3) | 11 (1.6) | 13 (1.9) |

| | | | | | | |
|---|---|---|---|---|---|---|
| Indus | 2103 | 0 | 0 | 308 (14.6) | 6 (1.9) | 11 (3.6) |

**Table 1. Channel network properties and errors of the recipes. Links compared % is fraction of total links for each CN. Disagree and expert errors %s are fractions of links compared.**

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
