# Peer review of "Determining flow directions in river channel networks using planform morphology and topology"

_Earth Surface Dynamics, 2019_

## Referee Comment (RC1) · Maarten Kleinhans (Referee) · 21 Jul 2019

This manuscript presents algorithms to improve river and delta channel networks derived from water/land-binarised and skeletonised imagery, specifically to assign flow directions to the links between channel nodes. The set of algorithms is tested against expert judgement and found to be accurate. As such the manuscript adds to a growing family of channel network production tools needed for graph-related and other network analysis tools. The paper is mainly method-oriented and presents no tests or explanations why certain algorithms were added to, or left out of, the workflow. There is some data analysis but very little discussion and comparison to work done in the literature.

[Figure]

While this can potentially be repaired, it requires doing such analyses and writing a paper about it, with much of the present manuscript in a supplement, suggesting rather major revisions.

In general the paper is very difficult to read as anything other than a recipe for accomplishing something, and what I would expect for this journal is emphasis on that something. For example, section 3.1 is very hard to read and is probably much better understood when graphically presented in schematics. It almost reads as a cooking recipe that tells the reader to add an ingredient without explaining why, and without explaining what would happen without it or with an alternative ingredient. Perhaps this can be resolved by much better figures that explain what the method does and what comes out of it, placing much of the present technical material in a supplement and focussing the paper on the science rather than the method. The evaluation can also be part of that, including explanatory reasons why those few links went wrong because that may tell us something very interesting about the method, why it works, and what basic understanding it embodies about fluvial systems. Or perhaps this manuscript is more suitable for another journal, if not a supplement to a paper about the science.

Detailed comments by figure and line numbers:

Text and figures are cluttered by abbreviations, many of which can be resolved. For instance DPA is unnecessary because the entire paper is about that thing so why not name the subalgorithms by the name that says what they do.

Figures are unclear and not so suitable in background choice for journal publication. This can easily be resolved.

Polish needed: there are multiple grammatical and spelling errors and figure panels need letters for reference in captions.

Fig1: black background is beautiful for presentation but make more readable white background for the paper. This also applies to some other figures.

Fig2: if in DPA_mdc colours between equation and schematics are supposed to correspond then something is wrong. The caption refers to the text for symbology, but readability would really improve if a figure explains that symbology. It says 'min dir change DPA_mdc' with different omega_ang in multiple places, but why and why these values cannot be understood from the figure. It says DPA everywhere so that is clearly redundant.

Fig 3: likewise, this fig is very very hard to read with all the unexplained symbology. Perhaps put in the supplement and make a fig for the paper that explains rather than technically records what the recipes do. For braided rivers the cycles are not connected to the rest, and that bit is the same as in the deltas so for clarity merge the two.

Fig4: again nice for presentation but as a figure it does not work. Why not make blue links for downstream and red for upstream with gray in between and white background.

Fig6: Nice results, but write out meaning of legend so it becomes readable

40 missing the most important problem here: bed slopes are nearly as much upward in downstream direction because of shoals and bifurcations, which requires a very different method to get the networks (Kleinhans et al. 2017, Van Dijk et al. 2019).

46,51 then why is there this remark in the online supplement readme that "Important: The Colville, Kolyma, Lena, Mackenzie, Yenisei, and Yukon channel network masks are not included in these Supplementary Data, as they were painstakingly created by Anastasia Piliouras"? What was so much work about it?

136 why this weight? this needs arguments and support. In Marra et al (cited in the paper) we tested and discuss a number of possibilities in view of fluvial morphodynamic functioning.

The link to the data https://doi.org/10.15485/1505624 leads to the repository but gives a blank page as result.

References

van Dijk,W.M., Hiatt, M. R., van der Werf, J. J., & Kleinhans, M. G. (2019). Effects of shoal margin collapses on the morphodynamics of a sandy estuary. Journal of Geophysical Research: Earth Surface, 124. https://doi.org/10.1029/2018JF004763

Note: this paper comes from a community where the authors are in alphabetical order. Willem Sonke is the lead author and did this work as part of his PhD thesis. Kleinhans, M., M. van Kreveld, Tim Ophelders, W. Sonke (lead author), B. Speckmann (PI), and K. Verbeek (2017). Computing Representative Networks for Braided Rivers. 33rd International Symposium on Computational Geometry (SoCG 2017), pp. 48:1-48:15. Editors: Boris Aronov and Matthew J. Katz. http://dx.doi.org/10.4230/LIPIcs.SoCG.2017.48

Maarten G. Kleinhans, 21 July 2019

---

## Referee Comment (RC2) · George Allen (Referee) · 10 Sep 2019

Schwenk et al. presents a method to assign steady-state flow directions to channel links of delta and braided rivers with complex river morphologies. This information is useful for a range of biogeochemical and hydrological flux processes. The manuscript is generally well written and I find the manuscript easy to understand. I think it was a high-quality study and I appreciate that the code and datasets were made freely available. The one major problem I found with this study is that, it might be submitted to the wrong journal. It is a methods study, essentially explaining the RivGraph Python package (https://github.com/jonschwenk/RivGraph), which may not be suitable for ESD, at

least according to the aims and scope of the journal. In my opinion this study is probably most appropriate for a journal like IEEE Geoscience and Remote Sensing Letters. However, that being said, if the Editor wishes to continue with the review process, I think Schwenk et al. is a nice contribution.

Major comments:

1. I think the manuscript would benefit from a paragraph in the introduction discussing other channel vectorization algorithms to provide additional context and motivation (e.g. RivaMap, RivWidth, RivWidthCloud, MERIT Hydro, etc.).

2. Include in the intro and/or abstract that RivGraph determines the steady-state, or mean long-term flow direction. Deltaic systems are often bidirectional flow and this point was only acknowledged in pass in the Conclusions.

3. I think a very large potential improvement of this approach would be automatic identification of inlets and outlets and this should either be implemented into the code or acknowledged in the "Improvements and Speed" section.

4. It appears that lakes and other non-channelized water bodies are not included in the Delta river masks. Were these removed? These features can be some of the most difficult to skeletonize and I am curious how the authors handled these features.

5. While the authors may have captured all the major sources of errors for their sample data set, applying these algorithms worldwide will likely cause a number of currently unidentified errors to be identified. I recommend noting this point somewhere in the manuscript main text (e.g. end of section 4.2).

Minor comments:

1. Add how the authors identified inlets and outlets. Was this done manually? Could it be automated?

2. Figures:

a. Panels should be in the same order as they are referred to in the main text.

b. Figures would benefit from having labeled panels (e.g. "a","b","c", etc..).

c. Figures are sometimes mislabeled (e.g. Figure 5 is referred to as Figure 6 several times).

d. Figures with maps: Add North arrow(s) to maps that are not oriented North as up.

3. If the authors wish to add an additional end-member sample, the braided section of the Congo River has a very distinct planform geomorphology and could be an additional case to test RivGraph. This idea is just a gentle suggestion, not a demand.

L76: Islands of size 20 pixels or less were removed (filled) from all channel networks. Please justify this action.

L87: Replace "GISs" with "GIS software packages"

L269: "the shortest link is selected as the one to be fixed (flipped), as DPAs are generally more certain about longer links." I probably don't completely understand but why not just flip the link with the lowest direction certainty?

L283: change "informations" to "information"

L421: "one second" Is this on one core or is this code paralllelized?

---

## Author Comment (AC1) · 17 Oct 2019

Please see the attached pdf.

Please also note the supplement to this comment:
https://www.earth-surf-dynam-discuss.net/esurf-2019-19/esurf-2019-19-AC1-supplement.pdf

―――――――――――――――――――――

---

## Author Response (AR1)

**Response to George Allen's comments**

(Responses italicized and blue.)

**Overall:**

Schwenk et al. presents a method to assign steady-state flow directions to channel links of delta and braided rivers with complex river morphologies. This information is useful for a range of biogeochemical and hydrological flux processes. The manuscript is generally well written and I find the manuscript easy to understand. I think it was a high-quality study and I appreciate that the code and datasets were made freely available. The one major problem I found with this study is that, it might be submitted to the wrong journal. It is a methods study, essentially explaining the RivGraph Python package (https://github.com/jonschwenk/RivGraph), which may not be suitable for ESD, at least according to the aims and scope of the journal. In my opinion this study is probably most appropriate for a journal like IEEE Geoscience and Remote Sensing Letters. However, that being said, if the Editor wishes to continue with the review process, I think Schwenk et al. is a nice contribution.

We thank George for the positive and constructive comments. Maarten Kleinhans (other reviewer) had the same concern: that this paper is too methods-y for Earth Surface Dynamics (ESD). We submitted this work to ESD because we believed that the ESD community is most likely to contain researchers who would find both the method and its implications interesting, although we acknowledge it is an atypical ESD submission. Accordingly, we corresponded with the Associate Editor both before and after reviews to ensure its suitability here and received a supportive response. We have also made additional efforts to appeal to a broader readership by contextualizing this research through comparisons with other work (L50-54, L60-62, L102-105), explaining the theoretical basis for DPAs when applicable (L109-119, L151-154, L170-173, L201-205, L218-223, L246-247, L253-255), and adding discussion about the implications of our results to process-form relationships across CNs (L21, L64-66, L70-71, L413-414, L437-444, L494-497).

We would also like to emphasize that this is not an overview of the RivGraph package. This manuscript represents only one piece of RivGraph, albeit one of the more complicated pieces. We intended this paper to be independent of RivGraph as the recipes described herein are only examples of possible recipes—optimized for the deltas and braided rivers we took as example cases, but hopefully generally applicable.

**Major comments:**

1. I think the manuscript would benefit from a paragraph in the introduction discussing other channel vectorization algorithms to provide additional context and motivation (e.g. RivaMap, RivWidth, RivWidthCloud, MERIT Hydro, etc.).

We appreciate the suggestion. As mentioned above, this paper is not intended to be a presentation of RivGraph, and therefore we prefer not to discuss in-detail the many tools available for pre-processing data. However, we do agree that it would be beneficial to the reader to be aware of available tools for preparing their channel network, so we have added some text mentioning these. We note, however, that (as far as we know), none of these tools "do it all," including resolving centerlines, network structure (links, nodes, and connectivity), and morphologic properties (width, length). L102-105.

2. Include in the intro and/or abstract that RivGraph determines the steady-state, or mean long-term flow direction. Deltaic systems are often bidirectional flow and this point was only acknowledged in pass in the Conclusions.

We have addressed this by adding a paragraph to the introduction. L36-42.

**3. I think a very large potential improvement of this approach would be automatic identification of inlets and outlets and this should either be implemented into the code or acknowledged in the "Improvements and Speed" section.**

This is a great suggestion and one we have spent considerable time achieving. For the methods described herein, however, we did not want to confound the thrust of this paper—automatically setting channel directions—with other pre-processing steps one may use to implement these techniques. For this same reason we have neglected to include methods of generating channel masks. However, the RivGraph package does indeed automatically find inlet and outlet nodes for braided rivers. For deltas, the lack of a linear overall flow direction (compared with a valley direction of braided rivers) and the wide range of possible delta configurations renders an automatic solution intractable. However, outlet nodes for deltas are automatically determined by providing a shoreline.

**4. It appears that lakes and other non-channelized water bodies are not included in the Delta river masks. Were these removed? These features can be some of the most difficult to skeletonize and I am curious how the authors handled these features.**

The methods of mask generation are described in the cited Piliouras and Rowland work. Separate masks were created for the channels and lakes, such that we could isolate the different types of water bodies for various metric calculations and inter-delta comparisons. This was done by classifying water bodies by size, such that the largest connected water body represents the channel network. We then resegmented the channel network by shape to remove lakes that were structurally connected to the channels. The non-channelized water bodies were therefore not 'removed' from the analysis in the present manuscript, but rather the masks presented here represent only the channel class. Future work includes plans to add structurally connected lakes to the network/topology, but as you point out, these features are difficult to skeletonize, and that is beyond the scope of this paper.

5. While the authors may have captured all the major sources of errors for their sample data set, applying these algorithms worldwide will likely cause a number of currently unidentified errors to be identified. I recommend noting this point somewhere in the manuscript main text (e.g. end of section) *We have added text explicitly mentioning that global application might require some recipe modification in the final paragraph of the conclusions. L502-503.*

**Minor comments:**

1. Add how the authors identified inlets and outlets. Was this done manually? Could it be automated? *Please see the response to Major comment 3.*

2. Figures:

a. Panels should be in the same order as they are referred to in the main text.

This has been fixed.

b. Figures would benefit from having labeled panels (e.g. "a","b","c", etc..).

**Done.**

c. Figures are sometimes mislabeled (e.g. Figure 5 is referred to as Figure 6 several times). *Corrected.*

d. Figures with maps: Add North arrow(s) to maps that are not oriented North as up. Added to Figure 1. We did not add arrows to Figure 4, as it is not intended to be a "map" figure, but rather display results.

3. If the authors wish to add an additional end-member sample, the braided section of the Congo River has a very distinct planform geomorphology and could be an additional case to test RivGraph. This idea is just a gentle suggestion, not a demand.

Thank you for the suggestion! Due to the relative similarity of large braided river morphology, we only analyzed two braided river CNs, but we expect very similar results for the Congo River as those achieved for the Brahmaputra and Indus Rivers.

**L76: Islands of size 20 pixels or less were removed (filled) from all channel networks. Please justify this action.**

We have added a line stating that this island-filling procedure is not strictly necessary, but reduces some of the noise by eliminating smaller channels that are relatively unimportant to the network topology. L90-91.

**L87: Replace "GISs" with "GIS software packages"**

*We ended up removing the reference to GIS in favor of citing specific community-developed tools.* **L102-105***.*

L269: "the shortest link is selected as the one to be fixed (flipped), as DPAs are generally more certain about longer links." I probably don't completely understand but why not just flip the link with the lowest direction certainty?

The problem is that while we have certainty estimates for various DPAs, we don't know their degree of certainty relative to each other. Because of this, we can't simply select the highest-certainty prediction. Instead, we resort to link length, as we have empirically observed that the final flow direction prediction is generally more reliable for longer links. Note that this only refers to (a very small number of) cases where there are multiple options for flipping links to fix an interior source/sink.

**L283: change "informations" to "information"**

Done.

**L421: "one second" Is this on one core or is this code parallelized?**

Added some text to clarify this is unparallelized and run on a typical desktop computer. As a side note, the recipes are generally not (easily) parallelizable because they require sequential setting of flow directions. However, the computational expense is minor enough that parallelization would not be needed, even if more complex recipes were created. L471.

**Response to Maarten Kleinhans's review**

Responses italicized and blue.

**Note: some comments have been rearranged in order to condense responses.**

This manuscript presents algorithms to improve river and delta channel networks derived from water/land-binarised and skeletonised imagery, specifically to assign flow directions to the links between channel nodes. The set of algorithms is tested against expert judgement and found to be accurate. As such the manuscript adds to a growing family of channel network production tools needed for graph-related and other network analysis tools.

The paper is mainly method-oriented and presents no tests or explanations why certain algorithms were added to, or left out of, the workflow. It almost reads as a cooking recipe that tells the reader to add an ingredient without explaining why, and without explaining what would happen without it or with an alternative ingredient. Perhaps this can be resolved by much better figures that explain what the method does and what comes out of it, placing much of the present technical material in a supplement and focusing the paper on the science rather than the method. While this can potentially be repaired, it requires doing such analyses and writing a paper about it, with much of the present manuscript in a supplement, suggesting rather major revisions. In general the paper is very difficult to read as anything other than a recipe for accomplishing something, and what I would expect for this journal is emphasis on that something.

The evaluation can also be part of [refocusing the manuscript], including explanatory reasons why those few links went wrong because that may tell us something very interesting about the method, why it works, and what basic understanding it embodies about fluvial systems. Or perhaps this manuscript is more suitable for another journal, if not a supplement to a paper about the science.

We thank Maarten for his review, and note that his concerns regarding the suitability of this paper for Earth Surface Dynamics (ESD) were shared by the other reviewer. This paper is atypical for ESD in that it is methods-oriented, but per our discussions with the handling editor is acceptable for publication here. However, we appreciate Maarten bringing to our attention the expectations of ESD readers and have attempted to give the paper more relevance among the ESD audience in three ways: First, we have added text to describe the reasoning behind each DPA while citing the relevant work supporting this reasoning (L109-119, L151-154, L170-173, L201-205, L218-223, L246-247, L253-255). Second, we have tried to contextualize this work better by better-defining the existing research landscape (L50-54, L60-62, L102-105). Third, we have tried to highlight the interesting process-form implications that arise from our study—particularly how the effectiveness of DPAs give us clues into the universality (or not) of particular morphodynamic features, and how the variability of the strength process-form relationships renders a morphology/topology-only approach difficult yet achievable (L21, L64-66, L70-71, L413-414, L437-444, L494-497).

There is some data analysis but very little discussion and comparison to work done in the literature. We have added relevant comparisons as described in the above reply. However, we are unaware of a similar approach/method in the literature to compare directly against. We are aware of directions being set for each link of a CN, but only manually (Marra et. al., Tejedor et al., etc.). Our comparison of the recipes against the expert decisions is essentially testing against these previous works, though not on the same CNs.

**For example, section 3.1 is very hard to read and is probably much better understood when graphically presented in schematics.**

We have reduced the clutter by eliminating redundant acronyms. We intend for readers to refer to Figure 2 while reading Section 3.1, as it does provide schematic illustrations of the DPAs.

**Detailed comments**

Text and figures are cluttered by abbreviations, many of which can be resolved. For instance DPA is unnecessary because the entire paper is about that thing so why not name the subalgorithms by the name that says what they do.

It says DPA everywhere so that is clearly redundant.

It is important to denote to readers that each DPA belongs to the class of all DPAs, hence the original notation. However, we have removed the "DPA" label from all text and figures and replaced it with the bold, italicized acronym for each DPA.

Figures are unclear and not so suitable in background choice for journal publication. This can easily be resolved.

Fig1: black background is beautiful for presentation but make more readable white background for the paper. This also applies to some other figures.

Fig4: again nice for presentation but as a figure it does not work. Why not make blue links for downstream and red for upstream with gray in between and white background.

We chose the black background for some figures for both practical and aesthetic reasons. The darkness of a black background provides a wider range of contrast that allows us to more clearly show lots of information (e.g. densely-packed flow directions for each link). We did try many color combinations on white backgrounds but found them all less clear than the black backgrounds. Please see Fig. 1 at the end of this document for some comparisons. We note that all figures in this manuscript comply with Earth Surface Dynamics figures guidelines (https://www.earth-surfacedynamics.net/for authors/manuscript preparation.html).

Polish needed: there are multiple grammatical and spelling errors and figure panels need letters for reference in captions.

We have re-edited and spellchecked the manuscript. Letters have been added to figure panels.

Fig2: if in DPA\_mdc colours between equation and schematics are supposed to correspond then something is wrong.

This is now fixed.

Fig 3. The caption refers to the text for symbology, but readability would really improve if a figure explains that symbology. It says 'min dir change DPA\_mdc' with different omega\_ang in multiple places, but why and why these values cannot be understood from the figure.

Fig 3: likewise, this fig is very very hard to read with all the unexplained symbology. Perhaps put in the supplement and make a fig for the paper that explains rather than technically records what the recipes do. For braided rivers the cycles are not connected to the rest, and that bit is the same as in the deltas so for clarity merge the two.

Unfortunately, the definitions of the thresholds ( $\omega$ ) are too involved to include in this figure. However, we have added two sentences to the caption to help explain their purpose and where to find their definitions. We explain in the text and the caption that recipes are combinations of DPAs with the purpose of setting all links' directions. The explanation for what the components (DPAs) do is given in detail in Section 3.1.

Fig6: Nice results, but write out meaning of legend so it becomes readable *The DPA abbreviations have been fully written in the revised caption.*

40 missing the most important problem here: bed slopes are nearly as much upward in downstream direction because of shoals and bifurcations, which requires a very different method to get the networks (Kleinhans et al. 2017, Van Dijk et al. 2019).

We were motivated by techniques that are globally-applicable, and high-resolution bathymetry is not widely available for most CNs (especially large ones). We have added text that clarifies our motivation and mentions the difficulties cited in this comment.

46,51 then why is there this remark in the online supplement readme that "Important: The Colville, Kolyma, Lena, Mackenzie, Yenisei, and Yukon channel network masks are not included in these Supplementary Data, as they were painstakingly created by Anastasia Piliouras"? What was so much work about it?

The devil's in the details. While it is now quite easy to create or obtain a binary channel mask, the quality of the mask can vary substantially, and the desired quality depends on the use. The adjective "painstakingly" was included to indicate that attention to detail was paramount in these masks' generation, and that it included a significant amount of fine-scale corrections. We have replaced this adjective with more precise terminology in the Supplementary readme.

136 why this weight? this needs arguments and support. In Marra et al (cited in the paper) we tested and discuss a number of possibilities in view of fluvial morphodynamic functioning.

Marra et al. used width, 1/length, and width/length as possible weights for computing topologic metrics. Here, we found that width alone was sufficient to define "main channels" and did not test other metric. This choice is now further explained in L201-205.

**The link to the data https://doi.org/10.15485/1505624 leads to the repository but gives a blank page as result.**

We are not sure why a blank page resulted—we are able to download the data from the provided DOI as of 9/24/2019.

**References**

van Dijk,W.M., Hiatt, M. R., van der Werf, J. J., & Kleinhans, M. G. (2019). Effects of shoal margin collapses on the morphodynamics of a sandy estuary. Journal of Geophysical Research: Earth Surface, 124. https://doi.org/10.1029/2018JF004763

Note: this paper comes from a community where the authors are in alphabetical order. Willem Sonke is the lead author and did this work as part of his PhD thesis.

Kleinhans, M., M. van Kreveld, Tim Ophelders, W. Sonke (lead author), B. Speckmann (PI), and K. Verbeek (2017). Computing Representative Networks for Braided Rivers. 33rd International Symposium

on Computational Geometry (SoCG 2017), pp. 48:1-48:15. Editors: Boris Aronov and Matthew J. Katz. http://dx.doi.org/10.4230/LIPIcs.SoCG.2017.48